# Endoplasmic Reticulum Stress Signaling and Neuronal Cell Death

**DOI:** 10.3390/ijms232315186

**Published:** 2022-12-02

**Authors:** Adalberto Merighi, Laura Lossi

**Affiliations:** Department of Veterinary Sciences, University of Turin, 10095 Grugliasco, Italy

**Keywords:** endoplasmic reticulum, endoplasmic reticulum stress, unfolded protein response, autophagy, apoptosis, cerebellar granule cells

## Abstract

Besides protein processing, the endoplasmic reticulum (ER) has several other functions such as lipid synthesis, the transfer of molecules to other cellular compartments, and the regulation of Ca^2+^ homeostasis. Before leaving the organelle, proteins must be folded and post-translationally modified. Protein folding and revision require molecular chaperones and a favorable ER environment. When in stressful situations, ER luminal conditions or chaperone capacity are altered, and the cell activates signaling cascades to restore a favorable folding environment triggering the so-called unfolded protein response (UPR) that can lead to autophagy to preserve cell integrity. However, when the UPR is disrupted or insufficient, cell death occurs. This review examines the links between UPR signaling, cell-protective responses, and death following ER stress with a particular focus on those mechanisms that operate in neurons.

## 1. Introduction

Among several cell organelles, the endoplasmic reticulum (ER) has been known since the mid of the last century, when, using electron microscopy, Porter and colleagues first observed it as a “fine lace-work stretching throughout the cytoplasm” in cultured chicken fibroblast-like cells [1]. A few years later, in 1954, Palade and Porter gave the ER its present name [2]. Today, we know that the organelle is visible as a membranous network of flattened discs and elongated tubules that cover a significant portion of the cell cytoplasm [3]. The ER membrane forms a barrier delimiting a discrete cellular compartment that prevents molecules from leaving or entering the cytoplasm while they are inside the ER lumen. Although the ER was once only known to play a role in protein synthesis, today it is widely accepted to also be involved in a wide range of other cellular processes. These include the intervention in lipid metabolism, the connections with cytoskeletal structures, and participation in cytoplasmic streaming. In addition, the ER is an important site for the storage of intracellular Ca^2+^, as store-operated Ca^2+^ entry (SOCE) is a ubiquitous Ca^2+^ signaling pathway activated by the Ca^2+^ depletion of the ER and various cellular stressors. In response to these stressors, reactive oxygen species (ROS) are frequently created and, remarkably, numerous cardiovascular, neurological, and immunological disorders are linked to changes in Ca^2+^ signaling and ROS generation [4].

Therefore, it is not at all surprising that the organelle is now seen as crucial in the control of the entire cellular metabolism. This important role played by the ER in all types of cells is even more crucial in the nervous tissue, where, broadly speaking, the ER regulation of proteostasis is essential for maintaining tissue physiology.

In the normal brain, the unfolded protein response (UPR) that follows ER stress is involved in the development, synaptic signaling/plasticity [5], and memory formation. In addition, the UPR plays a role in neurodegenerative diseases [6].

Among the several functions of the UPR in the physiology of the central nervous system (CNS), one can first mention the regulation of neurogenesis with direct impacts on the architecture and lamination of the cerebral and cerebellar cortices and the appearance of microcephaly when UPR is destroyed [7,8]. ER stress also intervenes in the genesis of dendritic spines and the assembly of glutamate receptors at synapses and, therefore, is linked to synaptic plasticity, long-term depression (LTD), and potentiation (LTP), a series of processes that are at the basis of memory formation [9,10,11,12].

In the pathological CNS, the main molecules involved in UPR have been proposed to play important roles in the genesis of Alzheimer’s disease, amyotrophic lateral sclerosis (ALS), schizophrenia, prion-mediated neurodegeneration, and several other brain dysfunctions and alterations in UPR signaling that negatively impact memory formation and behavior [6].

Although these observations mainly focus on the links between ER stress and neurons, recent research has shown that active UPR signaling is also biologically relevant in glia. For example, the mesencephalic astrocyte-derived neurotrophic factor (MANF), an ER-resident protein, has important functions in brain development given that its ablation led to a delay in the migration of cortical neurons together with impaired neurite outgrowth [13]. Under other conditions, activation of the UPR may cause astrocytes to take on an unfavorable phenotype: these cells release inflammatory mediators in response to ER stress, have less trophic support, and can spread ER stress to other cells [14].

Notably, several pathways of cell death in the brain have been associated with ER stress. Apoptosis, the best-characterized mode of programmed cell death (PCD), may be activated in neurons because of ER stress. The UPR following chronic unsolved ER stress can lead to apoptosis by various pathways involving c-Jun N-terminal kinase (JNK), glycogen synthase kinase 3/3β (GSK3/3β), CAAT/enhancer binding protein homologous protein (CHOP), or caspase-12 [15]. Neuronal apoptosis can also be triggered by alterations in ER Ca^2+^ homeostasis leading to several forms of neurodegeneration [16]. 

ER stress also affects autophagy, a mode of cell death mediated by lysosomes, as an unbalanced activation or inhibition of autophagy can disrupt proteostasis [17].

Here, we will discuss the link between ER stress signaling and cell death with a focus on central neurons. 

## 2. ER Morphology and Functions

### 2.1. Morphology of the ER

According to whether ribosomes are present or absent on the cytosolic aspect of the membrane, the ER is traditionally classified into rough ER (RER) and smooth ER (SER). Both spatially segregated and coupled compartments of the SER and RER are possible [3,18]. The ER can also be subdivided by taking into consideration the structure of its membrane. The nuclear envelope, sheet-like cisternae, and a polygonal array of tubules joined by three-way junctions are all included in this classification of the ER [19]. 

The ER is close to numerous other intracellular organelles and has a large track inside the cell that varies depending on the cell type. Among the others, the ER is associated with the mitochondria, which are particularly interesting to the present discussion as they play a pivotal role in apoptosis [20,21]. The association between the ER and the mitochondria occurs at the so-called mitochondrial-associated membranes (MAMs), which are essential for maintaining Ca^2+^ homeostasis [22]. Additionally, MAMs interact with the cell membrane for the maintenance of its stability and growth. This interaction is controlled by Ca^2+^ levels and by several proteins including stromal interaction molecule 1 in the ER and calcium release-activated calcium channel protein 1 in the cell membrane [23]. StAR-related Lipid Transfer Protein 3 (StARD3) and its close paralogue StARD3 N-terminal-like (StARD3NL) tether the ER and interact with endosomes [22]. Interestingly, the ER also appears to be involved in autophagy, as suggested by a link with the endolysosomal system [24]. The phagophore, one of the key structures in autophagy, extends and develops into a mature autophagosome when in contact with a specific ER assembly known as the omegasome [25,26].

### 2.2. Functions of the ER

#### 2.2.1. Protein Processing

The ER participates in the synthesis, folding, maturation, quality control, and destruction of secretory and transmembrane proteins [27]. It also makes sure that only properly folded proteins are transported to other cellular compartments or the cell membrane [28]. Approximately 30% of all proteins are cotranslationally directed to the ER lumen [29,30,31,32], where they are exposed to a setting rich in chaperones and foldases that aid in their folding, assembly, and post-translational modification. Thus, a properly folded conformation is the result of all protein-processing pathways carried out within the ER [29]. 

Protein misfolding is closely regulated since it may be harmful to the cell. It occurs continually but might get worse under unfavorable intrinsic and extrinsic events. To ensure that there are more opportunities to repair misfolded proteins or, if they are terminally misfolded, that they are disposed of by the cell, the ER operates the so-called ER-associated degradation (ERAD), a quality control mechanism that eliminates secretory proteins that have terminally misfolded [33,34].

#### 2.2.2. Lipid Synthesis

The ER is crucial for the maintenance of the various cellular membranes, the generation of lipid droplets and vesicles, and the buildup of fat for the storage of energy. Lipid production is restricted to organelle contact sites and membrane interfaces, and the lipid droplets/vesicles are expelled in a controlled manner. The ER adapts to shifting cellular lipid concentrations by dynamically modifying the shape of its membrane. Among the various molecules interacting with lipids housed in the ER are the sterol regulatory element-binding protein family of cholesterol sensors [35], and a series of enzymes that intervene in the metabolism of the cell membrane lipids [36]. These enzymes produce precursors that are later transformed into structural lipids, sterols, steroid hormones, bile acids, dolichols, prenyl donors, and a wide variety of isoprenoid species with crucial roles in cell metabolism [3,37].

#### 2.2.3. Transfer of Molecules from the ER to Other Cellular Compartments

The secretory route is primarily used to transfer the majority of proteins and lipids produced in the ER to other cellular components. The process is strictly controlled to maintain a constant metabolic flux, and deficiencies in secretion can have detrimental structural and functional effects on the ER. The formation of the coat protein complex II (COPII)-coated transport vesicles, named after the family of proteins that forms and covers them, is essential to this export process [38].

#### 2.2.4. Regulation of Ca^2+^ Homeostasis

In numerous intracellular and extracellular signaling networks, Ca^2+^ functions as a secondary messenger and is crucial for many cellular processes including cell death [39].

The ER, which serves as the primary cellular space for Ca^2+^ storage (see also Section 5. Adaptive reticulum stress and its relationship to Ca^2+^ dynamics), is critical in the management of Ca^2+^ levels, and many ER functions are in turn regulated by Ca^2+^ in a manner that affects the calcium balance of the entire cell [40]. For the ER to maintain a substantially higher physiological intraluminal Ca^2+^ concentration and oxidizing redox potential than the cytoplasm, both ER and cytosolic Ca^2+^ concentrations need to be tightly controlled by several intracellular mechanisms in close coordination with the cell membrane and mitochondria [41]. 

## 3. ER Stress

ER stress is a cellular state brought on by circumstances that interfere with ER homeostasis. The activation of adaptive mechanisms to deal with stress and regain ER equilibrium is part of the cellular response to ER stress. The perturbing substance(s) or circumstance(s), as well as the level and length of the stress, all affect this response [42]. 

There may be intrinsic and extrinsic factors affecting the ER equilibrium. Among the former are cell autonomous pathways that can be activated by neurodegenerative disorders, diabetes, and cancer. Among extrinsic factors, one can list cell environmental stress, exposure to ER stressors, enhancers of ER homeostasis, temperature, ROS production, and other distress [43].

### 3.1. ER Stress and the Unfolded Protein Response

Remarkably, a buildup of misfolded proteins in the brain is a hallmark of many neurodegenerative diseases [42]. UPRis activated in cells responding to ER stress. UPR reduces protein production, eliminates misfolded or unfolded proteins, and boosts the ability of the ER to fold proteins. Activating transcription factor 6 (ATF6), protein kinase RNA-activated (PKR)-like ER kinase (PERK), and inositol-requiring enzyme 1 (IRE1) are the three main sensors that primarily regulate the UPR [43] (Figure 1). The ER resident chaperone, heat shock protein A5 (heat shock protein family A (Hsp70) member 5, also known as glucose-regulated protein 78 (GRP78) and binding immunoglobulin protein (gene GRP78) (BiP)), normally binds the ER luminal domains of all three ER stress sensors, keeping them in an inactive state [44,45].

Misfolded protein buildup in the ER lumen activates BiP, releasing the three sensors. When IRE1 and PERK are freed from BiP, they trans-autophosphorylate and homodimerize or oligomerize to activate their corresponding downstream molecular pathway [44].

BiP dissociation from AFT6, however, reveals an ER export motif [45] that aids in its translocation to the Golgi apparatus [46]. According to this “competitive model” of UPR activation, BiP controls UPR signaling negatively. Other BiP-dependent or independent hypotheses, however, have been put forth to explain the mode(s) by which UPR signaling is regulated (reviewed in [47]).

### 3.2. Effects of UPR Activation

The activation of UPR leads to a series of cellular events comprising several first-response reactions such as the regulation of transcription and translation, and the activation of the protein degradation machinery. Overall, these strategies reduce the accumulation of incorrectly folded proteins in the ER, allowing equilibrium to be restored by adaptive and repair mechanisms. The UPR then turns off when the amount of incorrectly folded proteins decreases. Another important immediate effect of the UPR activation is the control of MAMs. 

Additionally, the activation of the UPR has a series of downstream consequences among which cell death is particularly relevant to the present discussion. Other consequences of the activation of UPR impact redox homeostasis, glucose, and lipid metabolism. These will not be discussed here for brevity, but readers will find a very interesting summary of the literature in [43]. 

The first-response reaction to UPR leads to several cellular processes among which are the regulation of transcription, translation, and miRNAs, protein degradation, and the control of MAMs. Subsequent downstream consequences have an impact on prosurvival mechanisms as well as activities such as cell division, proliferation, metabolism, and death. It is important to note that the cellular response changes from prosurvival to prodeath after the sustained activation of the UPR. These mechanisms have been recently extensively reviewed and we will only mention here those essential to understanding the links between ER stress and cell death.

Three primary UPR transcription factors, cytosolic ATF6 (ATF6f), XBP1, and ATF4 collectively initiate several adaptive responses to restore ER function and ensure cell viability along the three different UPR cellular branches. Each branch leads to transcriptional regulation [42], controlling the expression of many different genes that are crucial to cell functions and viability [48]. The IRE1-XBP1 pathway is implicated in an ER chaperone induction and ERAD capacity management [49], as well as cytoprotection promotion [50] and miRNA cleavage that controls cell death-inducing caspases [51]. On the other hand, autophagy genes that support survival under ER stress have their transcription levels increased by ATF4 [52].

It has also been demonstrated that ER stress can control the apoptosis execution phase by temporarily increasing the levels of proteins that inhibit apoptosis (IAPs). According to several authors, ER stress induces the expression of the cellular inhibitors of apoptosis protein 1 (cIAP1) and 2 (cIAP2) and the X-linked inhibitor of apoptosis protein (XIAP). This induction is crucial for cell survival since it postpones the initiation of caspase activation and death. The production of cIAPs by PERK and the temporary activity of PI3K-AKT signaling indicate that PERK actively suppresses the ER stress-induced apoptotic program in addition to facilitating adaptation to ER stress [53].

Following ER stress, components of the UPR primarily activate two distinct protein degradation pathways: ubiquitin-proteasome-mediated degradation via ERAD and lysosome-mediated protein degradation via autophagy. Autophagy is activated as a secondary reaction to control protein build-up when the accumulation of misfolded proteins overwhelms ERAD [50,54]. ER stress has been shown to activate autophagy in mammalian systems through the PERK (eIF2α) and IRE1 (TRAF2/JNK) branches of the UPR [55]. Beclin 1, a significant participant and autophagy regulator, is activated by IRE1-JNK signaling through the phosphorylation of BCL-2 and subsequent dissociation from Beclin 1. As a result, the ATG proteins necessary for the development of the autophagolysosome are subsequently activated [56].

Numerous data in the literature highlight the significance of ER-mitochondria communication in controlling ER homeostasis and coordinating cellular responses to ER stress, which can either restore cellular balance or result in cell death. MAMs are primarily in charge of maintaining Ca^2+^ homeostasis and lipid transport [44]. This regulates mitochondrial metabolism and apoptosis [57].

Bax-inhibitor-1 (BI-1) is one of the MAM proteins that control mitochondrial Ca^2+^ uptake and apoptosis. In BI-1-defective cells, there is IRE1 hyperactivation and elevated levels of its downstream targets because BI-1 is a negative regulator of IRE1-XBP1 signaling [58]. When the UPR activates apoptosis, the mitochondrial membrane becomes permeable, and the ensuing Ca^2+^ transfer may cause the release of cytochrome c from the mitochondria [59]. The interactions of the mitochondria with the ER during sublethal ER stress are less clear. In contrast to deadly levels of ER stress, the latter causes more ER-mitochondria connections, allowing for Ca^2+^ transfer and enhancing ATP synthesis through increased mitochondrial metabolism [60].

### 3.3. ER Stress in Neurodegenerative Diseases

As mentioned previously, the UPR plays a major role in several neurodegenerative diseases. The importance of ER stress and UPR under these conditions has recently been reviewed [61]. We will briefly mention here the most relevant data in the literature demonstrating an intervention of the three branches of the UPR in the most important neurodegenerative conditions. 

After neurotoxin administration or in α-synuclein Tg mice to model Parkinson’s disease, all the branches of the UPR were implicated. Blocking PERK signaling with salubrinal, a specific inhibitor of eIF2α, or by knocking out CHOP, resulted in neuroprotection [62,63], viral transfection of XBP1 reduced the loss of dopaminergic neurons [64] whereas knocking out ATF6 exacerbated neurodegeneration [65]. 

PERK signaling was shown to have a primary role in amyotrophic lateral sclerosis (ALS) using the SOD1 transgenic mouse model after targeting the pathway with different pharmacological or DNA-modifying approaches, leading, according to the experimental conditions, to an aggravation (enhanced SOD1 aggregation) or an amelioration (extended life span) of the ALS phenotype [66]. IRE1 signaling was also implicated after the conditional knockout of XBP1 [66,67].

PERK signaling and IRE1 signaling were also involved in Alzheimer’s pathology. The intervention of the PERK pathway was demonstrated using several strains of transgenics and different pharmacological manipulations leading to the demonstration that when the pathway is blocked there is global neuroprotection [68,69,70], reduced tau phosphorylation [71], improved learning and memory, and LTP [72]. However, targeting protein translation did not ameliorate spatial learning and memory deficits in the hAPP-J20 mouse model of Alzheimer’s disease [73]. When IRE1 signaling was blocked in conditional knockouts of APP/PSEN1 mice, there was a reduced accumulation of amyloid β and an amelioration of functional and cognitive responses [74,75] and after the associated adenoviral transfection of XBP1, there was a behavioral improvement with an amelioration of the histological phenotype [76].

In the mutant Htt transgenic mouse model of Huntington’s disease, knocking out ATF4 has no effects on Htt aggregation [77] whereas targeting the IRE1 signaling pathways resulted in an improvement in motor performances and reduced Htt accumulation [77,78].

Again, after studies on mouse models, the UPR was also implicated in Charcot–Marie–Tooth disease, Pelizaeus–Merzbacher disease, and prion-related disease [61].

## 4. Death Modes Activated by ER Stress

As mentioned in the Introduction, ER stress can trigger several different modes of cell death. We will consider here those that are particularly relevant to our discussion. 

### 4.1. Apoptosis

Among the different forms of PCD, apoptosis is by far the most widely investigated as its discovery and characterization date back to the 1850s. Apoptosis can be triggered by two main different mechanisms, referred to as intrinsic (mitochondrial) and extrinsic (death receptor-mediated) [21]. It is worth noting that unresolved ER stress can activate both pathways and all three of the UPR’s branches are involved in both types of apoptosis (Figure 1). As one of us has very recently reviewed these mechanisms [21], we will here only mention the most relevant to the present discussion. 

#### 4.1.1. General Concepts

A collection of ultrastructural characteristics that appeared following a specific sequence after TEM observation was the basis for the initial description of apoptosis as a separate type of cell death. Cell fragmentation, cytoplasmic condensation, and phagocytosis are the steps in the sequence [79,80]. In the beginning, it is normal to see chromatin condense and segregate into sharply defined crescent-shaped masses in contact with the nuclear envelope. The chromatin masses are extraordinarily electron-dense because they are composed of tiny, granular material that is closely packed. This initial condensation of the nucleus is capable of leading to true nuclear pyknosis. Along with nuclear changes, the cytoplasm condenses, causing the cell membrane to become convoluted and the growth of protrusions of various sizes, giving the cell a star-like appearance. In the later stages, the cytoplasm density increases, and some vacuoles may form, but the cellular organelles are undamaged, even though the loss of cytosol causes them to become exceptionally tightly packed. As the process evolves, ribosomes are no longer bound to the RER, and the cell and its nucleus take on a more atypical appearance, with nuclear fragmentation in small masses surrounded by an intact envelope. In tissues, apoptotic bodies with condensed chromatin and intact organelles are eventually observed before being swiftly removed by macrophages or neighboring cells and digested within heterophagosomes. Efferocytosis is the name frequently given to this process [81].

As mentioned, apoptosis can take one of two forms: intrinsic or extrinsic [82]. The former is a type of controlled cell death that is triggered by changes to the intracellular microenvironment, whereas the latter is induced by changes to the extracellular microenvironment. Despite sharing a similar morphology, the two exhibit distinct metabolic activation processes. Unless cells have phagocytic capabilities, end-stage apoptosis in vitro is often followed by a complete collapse of the plasma membrane and the acquisition of a necrotic morphology (secondary necrosis) [83]. Recent studies demonstrated that secondary necrosis, also known as pyroptotic cell death, depends on the cleavage of DFNA5 by CASP3 during apoptosis with the creation of holes in the cellular membrane [84].

Research on the fruit fly *Drosophila melanogaster* and the nematode worm *Caenorhabditis elegans* has provided the majority of our knowledge regarding the gene regulation of apoptosis [85,86,87,88,89]. When the *C. elegans* death machinery genes’ orthologs in humans and *D. melanogaster* were cloned, it became clear that these genes function similarly in regulating apoptosis across all previously researched species.

#### 4.1.2. Caspases

Caspases are a class of proteases (caspase = cysteine aspartases) that include *C. elegans* homologs in mammals [90]. The first mammalian caspase to be identified was CASP1 (ICE) [91,92]. Fourteen members of the caspase family were then divided into three subfamilies based on the peptide sequence preferences of their substrates: (i) the ICE-protease subfamily of inflammatory caspases (CASP1, 4, 5, 13, and 14, as well as the murine CASP11 and 12); (ii) the CED-3 subfamily of apoptotic effector caspases (CASP3, 6, and 7); and (iii) the CASP2 subfamily. Effector caspases have a short pro-domain and destroy critical cellular proteins [90]. The other caspases of the family mediateprotein–protein interactions and may only induce apoptosis under specific conditions (perhaps except CASP1 and 11). 

Each caspase is synthesized as a zymogen that must be converted into an active enzyme at specific cleavage sites [93]. The initial caspases activate more downstream caspases, which, in turn, trigger a proteolytic cascade that ends with the execution of apoptosis.

Different caspase subgroups are activated in response to pro-apoptotic stimuli. Extrinsic apoptosis is caused by FasR/TNF-induced cell death, which is mediated by CASP3, 6, and 8, whereas intrinsic apoptosis is caused by mitochondria-associated cell death, which is mediated by CASP9, 3, Apaf1, and cytochrome c [94]. These two routes are not entirely independent because a link was discovered through BID, a protein that controls cytochrome c release from mitochondria in response to the activation of cell surface death receptors [95,96].

The apoptosome is a large protein complex that plays a major part in apoptosis [97,98]. The activation of the apoptosome depends on specific scaffolding proteins. Biochemical research has demonstrated that cytochrome c, Apaf1, and CASP9 are also necessary for the cleavage of CASP3 [99,100]. When Apaf1 binds to cytochrome c (also known as Apaf2) after it has been released from mitochondria at the start of apoptosis, a series of conformational changes take place, leading to Apaf1 multimerization and association with proCASP9 (also known as Apaf3), resulting in the formation of the approximately 1 MDa molecular weight apoptosome [101].

Early research using 293T cells from human embryonic kidneys revealed that BCL-XL binds with CASP9 and Apaf1 to prevent CASP9 maturation, which is controlled by Apaf-1, a process that was thought to be evolutionarily conserved from nematodes to humans at the time [102,103]. The BCL-2 family members, however, were shown not to interact with Apaf1 in later investigations [104,105,106].

#### 4.1.3. BCL-2 Proteins

At least fifteen proteins were identified and categorized into the so-called BCL-2 protein family [97,107,108,109,110] after the BCL-2 protein was discovered in mammals [111,112]. Recent reviews of the literature on the biology of these proteins can be found in [21,113,114,115].

Every member of the family possesses at least one of the four conserved BH domains (BH1, BH2, BH3, and BH4). BCL-XL and BCL-W are two of them that are survival factors, whereas BAX, BAK, BAD, and BID are pro-apoptotic factors. Cells are prevented from dying by members of the BCL-2 family that inhibit apoptosis. Like the membrane-inserting domains of colicins and diphtheria toxin, BAX and BAK can produce pores that serve as channels for ions or even proteins to traverse the outer mitochondrial membrane [116]. Inhibiting the pore-forming activity of BAX and BAK can block the release of cytochrome c from mitochondria, prevent the assembly of the apoptosome, and spare cells from death when BCL-2 and BCL-XL are located on the cytoplasmic side of the outer mitochondrial membrane [95,96]. It is worth noting that BCL-2 and BCL-XL bind with Apaf1 but are unable to prevent CASP9 activation in mammals [106].

#### 4.1.4. Apoptotic Pathways

The two main apoptotic pathways can be distinguished by their relative timing of caspase activation and the release of mitochondrial cytochrome c. While in extrinsic apoptosis an effector caspase is triggered before mitochondrial alterations, in intrinsic apoptosis, cytochrome c is liberated from the mitochondrial intermembrane gap before caspases are activated. 

The first step in the intrinsic pathway is the activation of intracellular sensors that identify, e.g., DNA damage, viral infections, or the absence of survival signals from neighboring cells. On the other hand, the extrinsic pathway starts with an extrinsic pro-death signal [117]. 

The two pathways should not be thought of as separate or competing with one another. This is not unexpected because they converge on the mitochondria, which serve as cross-talk organelles that link one to the other. Both pathways eventually converge on the execution phase, the last stage of apoptosis [21].

##### Intrinsic Apoptosis

The mitochondrial pathway of apoptosis is another name for intrinsic apoptosis. Intrinsic apoptosis is described as *a kind of controlled cell death initiated by disturbances of the extracellular or intracellular milieu, marked by mitochondrial outer membrane permeabilization, and precipitated by executioner caspases, primarily CAS*P3 [82]. Interestingly, and in contrast to extrinsic apoptosis, intrinsic apoptosis may be triggered by multiple mechanisms. A few examples of its causal events are, in addition to ER stress, growth factor depletion, DNA damage, ROS burden, replication stress, microtubular alterations, and mitotic abnormalities [118,119,120,121,122,123,124].

One key stage in intrinsic apoptosis is the release of cytochrome c into the cytoplasm. A rupture of the mitochondrial outer membrane and/or the so-called mitochondrial permeability transition (MPT) is regulated by the permeability transition (PT) pore, a voltage- and Ca^2+^-sensitive pore [125]. As a consequence, substantial and irreversible mitochondrial outer membrane permeabilization (MOMP) arises [126]. 

Members of the BCL-2 protein family tightly regulate MOMP. The intrinsic apoptotic pathway’s initial stage involves the interaction of pro- and anti-apoptotic BCL-2 family members. They can be divided into three groups based on structural and functional characteristics [114]. The first group is the BH3 proteins, which include the activators and sensitizers BAD, BID, BIK, BIM, BMF, HRK, NOXA, and p53 upregulated modulator of apoptosis (PUMA). These detect cellular damage and, either directly or indirectly, activate the second group’s members. The executioner members of the second group, BAX, BAK, or BOK, permeabilize the mitochondrial outer membrane and let molecules from the intermembrane space of the mitochondria escape into the cytosol. The third category consists of proteins such as BCL-2, which suppress both BH3 activators and executioner proteins to operate as anti-apoptotic agents.

Pro-apoptotic BCL-2 family members were found in the cytosol or on the cytoplasmic surface of the mitochondrial outer membrane, but anti-apoptotic family members were only found on the latter, according to initial subcellular localization studies [94]. Later, BCL-2-related proteins were found in the RER, the Golgi apparatus, the nucleus, and the peroxisomes, among other intracellular organelles. It thus became clear that BCL-2 protein distribution inside the cell is a dynamic process that is greatly influenced by changes in the cellular microenvironment [127,128]. BCL-2 family proteins regulate MOMP remotely at the level of these organelles, but they also play a role in fundamental cellular processes other than apoptosis such as calcium homeostasis, cell cycle control, and cell migration. However, we can concentrate on mitochondrial localization for the anti-apoptotic family members when focused on apoptosis.

Numerous in vitro and/or in vivo studies have proven that the BCL-2 family members are essential for controlling MOMP and that the intrinsic apoptotic machinery has striking overlaps under the numerous circumstances in which it may be triggered. They have also shown that the BCL-2 family of proteins must be active for virtually all nucleated cells to survive. Pro-apoptotic gene disruption in mice caused cell death in particular regions. For instance, BCL-2, BCL-XL, and BCL-W were needed for erythropoiesis, neurogenesis, and spermatogenesis, respectively [129]. Several strains of mice that have had certain pro- or anti-apoptotic BCL-2 family proteins genetically silenced succumb to early death or exhibit severe developmental abnormalities. It has been shown that the co-deletion of Bax, Bak1 (and Bok) makes many different types of cells significantly more resistant to different death triggers [130]. However, in vivo, such deletion is detrimental to survival as shown by newborn mortality in mice due to severe nervous system and hematological system developmental defects [131]. The important reduction in immature neuron apoptosis in BCL-XL+BAX-deficient mice [132], the postnatal diminished susceptibility to neuronal apoptosis in BCL-XL overexpressing transgenics [133], the reduction in neuronal survival in bcl-2 deficient mice [134,135], or the diminution of neuronal loss during the aging process are examples of the negative effects of changing the physiological balance between the pro- and anti-apoptotic members of the family. 

As the preceding description suggests, the interactions of the three groups of BCL-2 family members are quite complex, and at least three models, the direct activation model, the displacement model, and the unified model, have been put forth to account for these interactions [136]. They are discussed in depth in [21].

When activated, MOMP causes the release of apoptogenic factors from the mitochondrial intermembrane space, the two most important of which are cytochrome c and SMAC/DIABLO [137]. The two proteins are released into the cytosol thanks to the modification of the mitochondrial cristae [138]. As mentioned, the cytosolic pool of cytochrome c binds to APAF1 and proCASP9 to assemble the apoptosome, which is in charge of activating CASP9 [100]. CASP9, in turn, catalyzes the proteolytic activation of the executioner CASP3 and 7 when active. 

After entering the cytosol and undergoing a proteolytic maturation process that releases its latent IAP-binding domain, SMAC connects to members of the IAP protein family, including XIAP, to induce apoptosis [139]. By physically affixing to and blocking the caspases, XIAP is the only member of the IAP protein family that can prevent apoptosis [140]. Other IAP proteins, such as c-IAP1 and c-IAP2, also negatively regulate apoptosis, however, by different mechanisms, such as upregulating the potent anti-apoptotic factors CASP8 and c-FLIP [141], inactivating caspases through their E3 ubiquitin ligase activity [142,143], ubiquitinating RIPK1 in conjunction with pro-survival NF-KB signaling [144].

Evidence suggests that executioner caspases may not necessarily cause intrinsic apoptosis to occur when a previously ill-defined point-of-no-return is crossed [145]. In line with this theory, inhibiting post-mitochondrial caspase activation through genetic means or with particular pharmacological inhibitors frequently causes a delay in intrinsic apoptosis (but does not always stop it), as it promotes the transition to other types of cell death [145]. Additionally, when a small number of mitochondria are affected by an increase in MOMP, the ensuing caspase activation is sublethal rather than fatal and encourages genomic instability rather than cell death [146]. Overall, the aforementioned studies suggest that CASP3 and CASP7 are more likely to facilitate intrinsic apoptosis than to be crucial to it. As a result of the respiratory restriction brought on by the loss of cytochrome c, it is important to keep in mind that MOMP eventually results in the dissipation of the mitochondrial transmembrane potential, which has disastrous effects on cell survival. 

##### Extrinsic Apoptosis

A type of controlled cell death known as extrinsic apoptosis is brought on by modifications in the extracellular environment [82]. *Death is in this case triggered by extrinsic stimuli that can activate a very small number of structurally similar receptors*, in contrast to intrinsic apoptosis. Extrinsic apoptosis is generated by two different types of plasma membrane receptors: death receptors, which are activated by the binding of their cognate ligand(s), and dependence receptors, which are activated when the levels of their particular ligands drop below a predetermined threshold [147,148]

Death receptors have an intracellular “death domain” that is approximately 80 amino acids long. The death domain acts as a docking site for other pro-apoptotic proteins such as FADD, resulting in the formation of a membrane-bound DISC and the subsequent activation of caspases. The better-characterized death receptors are FasR, TNFR1, TRAILR1, TRAILR2, TNFRSF25, TNFRSF21, EDAR [149], and p75NTR [148,150,151]. Death receptors engage with their particular ligands at the cell surface to entice the recruitment of the adaptor proteins FADD and TRADD and downstream CASP8 [152].

Based on an evolutionary study, Bridgham and colleagues divided death receptors into two classes [153]. Members of the first class include EDAR, p75NTR, and TNFRSF21, which are primarily involved in developmental processes. Members of the second include FasR, TNFR1, TRAILR1, TRAILR2, and TNFRSF25, which play a significant role in cell death. For a recent review of the main features of these receptors, see [21]. 

##### Regulation of CASP8 Activity and DISC Formation

Extrinsic apoptosis is dependent on the development of DISC, a dynamic complex of many proteins at the intracellular tail of the death receptor that always contains FADD and CASP8. DISC controls the activity of CASP8, or CASP10 in some circumstances, in conjunction with the mitochondrial respiratory chain protein complexes I and II [154,155]. 

DISC assembly changes based on the death receptor that the apoptotic signal activates. In the case of FasR and TRAILRs, the linked ligands, FasL and TRAIL, respectively, stabilize the homotrimers of the prepared receptors to cause a conformational shift at their intracellular tails that facilitates the death domain-dependent connection of the adaptor FADD [156]. In turn, FADD promotes the death effector domain-dependent recruitment of CASP8 (or CASP10) and various isoforms of c-FLIP, which, in turn, support for the DISC assembly. Thus, these death receptors directly interact with FADD. On the other hand, because TRADD is first activated before recruiting FADD, its role in the instance of TNFR1 ligation is indirect. In this case, TRADD acts as an adaptor for complex I assembly [157].

Numerous studies have been conducted on the molecular processes that regulate CASP8 activity in response to the stimulation of the death receptor. The maturation of the caspase is caused by a series of events that are triggered by the binding of CASP8 to FADD at the DISC. Depending on their death effector domains, this enables CASP8 molecules to form a linear filament that promotes homodimerization and subsequent autoproteolytic cleavage [158,159]. A crucial role is played in this situation by c-FLIP, a close relative of CASP8 that lacks catalytic activity [160]. Notably, CASP8 is inhibited and activated by c-FLIPS and c-FLIPL, respectively, by altering CASP8 oligomerization [161,162]. The concept that elevated c-FLIPL expression prevents rather than stimulates extrinsic apoptosis is supported by the observation that CASP8 and c-FLIP isoforms are recruited at equivalent levels at the DISC [163]. Notably, NF-KB regulates the transcription of the gene CFLAR, which codes for c-FLIP, and which, under certain circumstances, primarily contributes to pro-survival TNFR1 signaling [160]. Additional post-translational processes, such as FasR activation, PLK3-catalyzed phosphorylation, and deubiquitination, appear to control CASP8 enzymatic activity [82].

##### The Execution Phase of Apoptosis

The execution phase consists of a succession of events directed by the so-called executioner caspases and is a point of convergence of the intrinsic and extrinsic apoptotic pathways. Executioner caspases, of which CASP3 is the most significant, catalyze a number of the structural and biochemical events of apoptosis, including DNA fragmentation [164], the exposure of phosphatidyl serine (PS) at the cell membrane [165], and the formation of apoptotic bodies [166,167]. By accelerating the proteolytic inactivation of ICAD and enabling CAD to restart its enzymatic activity, CASP3 promotes DNA fragmentation [168]. CASP3 also controls PS internalization and externalization. By activating phospholipid scramblases, which promote PS externalization [169,170], or by inactivating phospholipid flippases, which promote PS internalization [171], the protease increases PS exposure at the cell membrane. In response to apoptotic stimuli, cleaved (active) CASP3 promotes PS exposure via at least two distinct pathways. In one case, PS externalization is caused by the cleavage of XKR8 by CASP3, which then interfaces with BSG or NPTN to form a complex that scrambles phospholipids [170]. In the second, ATP11A and ATP11C’s flippase activity is inhibited by CASP3 [171].

Interestingly, however, PS exposure is not always linked to intrinsic or extrinsic apoptosis because it is a cell-type-specific event and may also be related to non-apoptotic cell death in necroptosis and efferocytosis [172].

Death receptors can cause extrinsic apoptosis in one of two ways: in some cells (type I), this process is independent, whereas, in other cells (type II), it is dependent on mitochondria [173,174]. The overexpression of anti-apoptotic BCL-2 proteins, the simultaneous deletion of Bax and Bak1, or the absence of BID did not affect the CASP8-dependent proteolytic maturation of executioner CASP3 and 7 in, for example, thymocytes and mature lymphocytes (type I cells) [174,175]. However, extrinsic apoptosis necessitates the proteolytic cleavage of BID by CASP8 in, for example, hepatocytes, pancreatic cells, neurons, and the majority of cancer cells (type II cells), in which XIAP inhibits CASP3 and 7 activations [176,177]. Survinin, an unusual member of the IAP protein family, has been shown to control the basal levels of CASP3 activity in slice-cultivated cerebellar granule cells (CGCs) [178].

### 4.2. Pathways Linking Apoptosis and ER Stress

After the death receptors bind their ligands, the activated proapoptotic molecules become transcriptionally upregulated in response to ER stress, which releases cytochrome c. The UPR’s IRE1 and PERK arms have both been connected to the induction of apoptosis during ER stress (Figure 1). In particular, the regulation of apoptosis during ER stress has been linked to CHOP, a transcription factor that is downstream of PERK and a direct target of ATF4. CHOP-induced expression of GADD34 causes the dephosphorylation of p-eIF2α, which reverses translational inhibition and permits the transcription of the genes involved in apoptosis [179].

While suppressing the transcription of some anti-apoptotic BCL-2 family members such as MCL-1, CHOP promotes the transcription of BIM and PUMA [180]. Additionally, the ATF4/CHOP pathway has been shown to boost the expression of other proapoptotic genes, including some of those that support extrinsic apoptosis, e.g., TRAILR1 and TRAILR2 [180]. In addition to CHOP, p53 has also been hypothesized to play a role in the ER stress-induced direct transcriptional upregulation of BH3-only proteins [181].

Although IRE1-XBP1 signaling mostly favors survival, IRE1 can also induce apoptosis. Apoptosis signal-regulating kinase 1 (ASK1) and its downstream substrates JNK and p38 MAPK can be activated by IRE1 through direct interaction with TRAF2 [182,183]. Several BCL-2 family members are regulated by JNK phosphorylation. As a result, there is the activation of proapoptotic BID and BIM and the suppression of antiapoptotic BCL-2, BCL-XL, and MCL-1 [184,185]. Additionally, p38 MAPK phosphorylates and activates CHOP, which, in turn, promotes apoptosis by increasing the expression of BIM and TRAIL2 [186,187]. It was intriguingly suggested that ER stress and MAPK signaling have a positive feed-forward connection, with ER stress-induced MAPK signaling, which then results in an increase in ER stress [188]. Through protracted RIDD activity, which breaks down the mRNA of protein-folding mediators, IRE1 signaling may also help induce apoptosis [189].

Interesting research suggests that miRNAs play a role in the onset of apoptosis after sustained ER stress. For instance, miRNA29a, which is generated during ER stress via ATF4, causes the antiapoptotic BCL-2 family protein MCL-1 to be downregulated and so promotes apoptosis [190]. Additionally, miRNA7 has been associated with ER stress-induced apoptosis. IRE1 decreases miRNA7 levels, promoting the stability of RNF183, a membrane-spanning RING finger protein. The anti-apoptotic BCL-2 family member BCL-XL is ubiquitinated and subsequently degraded as a result of the RNF183 E3 ligase domain. Increased apoptosis thus results from increased RNF183 expression via IRE1 during sustained ER stress [191].

### 4.3. Necrosis and Necroptosis

Necrosis is the death of a cell or group of cells, usually connected to one or more outside factors that result in non-physiological cell death. Necrosis can be broken down into two stages: the cell’s initial degradation and its death as a result of its failure to maintain homeostasis [192]. In general, necrosis involves a large number of cells and is associated with an inflammatory response. In irreversibly damaged cells, chromatin condensation without major alterations in its distribution is one of the morphological changes indicative of this type of cell death. Normal chromatin condensation results in small clusters with amorphous borders that are poorly delineated by the nucleoplasm around them. Mitochondria exhibit shape alterations, matrix modifications, and membrane rupture. Later, phases include the loss of chromatin and the disintegration of cell organelles and membranes. Mononuclear phagocytic cells finally remove necrotic material from tissues [193].

Necroptosis, a specific mode of cell death with mixed features of necrosis and apoptosis, differs from typical necrosis, which is governed by several signal transduction pathways and is a process of positive energy consumption, despite sharing comparable physical traits. Necroptosis controls some cellular signals to ensure that cells are not fully inactive when exposed to external trauma, hence minimizing damage [194,195].

Two crucial conditions must be met for necroptosis to take place: cells must express RIPK3 and CASP8 activity must be inhibited [196]. Death receptors including TNFR, TLR3/4, and INFR are the main causes of necroptosis. When CASP8 is inactive or lost, RIPK1 is activated. This, in turn, activates RIPK3 which increases the phosphorylation of MLKL. MLKL then connects with FADD to generate the necrosome, i.e., a molecular complex made of RIP1, RIP3, and FADD [197]. 

The inactivation of caspases can transform apoptosis into bona fide necrosis or necroptosis [198]. 

### 4.4. Pathways Linking Necroptosis and ER Stress

Necroptosis is induced after ER stress in a mouse model of spinal cord injury that occurs in vivo, and the location of MLKL and RIPK3 on the ER in necroptotic microglia and macrophages suggests a connection between necroptosis and ER stress in these cells [199]. Another example of the connection between ER stress and necroptosis was described in L929 murine fibrosarcoma cells where tunicamycin induces necroptosis after the activation of TNFR1 [200].

### 4.5. Autophagy

Autophagy is a kind of cell death in which lysosomal enzymes actively ingest the cytoplasm before the appearance of nuclear changes. The most thoroughly researched caspase-independent form of cell death has been autophagic cell death [201,202], however, the role of lysosomes in cell death has been intensively researched for a very long time before autophagy was recognized as a specific form of death [80,192,203]. One of the most distinctive features of autophagic cell death is the creation of large lysosomal-derived vacuoles in the cytoplasm. Autophagic cells eventually exhibit many of the hallmarks of apoptosis, but many of them do so noticeably later and before the expected nuclear changes of apoptosis become apparent. When the cytoplasm has been damaged to around 75 percent of its original volume, it starts to condense, exposing chromatin margination and coalescence. During this time, the cell’s remains undergo conventional apoptosis-like phagocytosis [201].

### 4.6. Pathways Linking Autophagy and ER Stress

Autophagic cell death has also been linked to ER stress. When apoptosis is experimentally inhibited or in response to treatments that specifically cause caspase-independent autophagic cell death, autophagy not only helps cells to survive but can also mediate nonapoptotic cell death [55].

Autophagy is facilitated through IRE1a-mediated TRAF2 and ASK1 recruitment, followed by JNK activation. Beclin-1 is released as a result of the JNK-mediated phosphorylation of BCL-2 (although XBP1s also transcriptionally upregulates its expression), and it then interacts with the ULK1 complex to enhance vesicle nucleation, which, in turn, causes the autophagosome to form [204].

By causing vesicle elongation, activated PERK can trigger autophagy through ATF4, whereas Ca^2+^ release from the ER lumen via the inositol 1,4,5-trisphosphate (IP3) receptors (IP3Rs) can relax mTOR inhibition on the ULK1 complex [50].

### 4.7. Ferroptosis

Ferroptosis is an iron-dependent form of PCD [205], which is distinct from other forms of cell death such as apoptosis and necrosis, is fueled by iron-dependent phospholipid peroxidation, and is characterized by the accumulation of lipid peroxides due to the failure of glutathione-dependent antioxidant defenses [206]. Several cellular metabolic pathways control ferroptosis, including ER stress, redox homeostasis, iron handling, mitochondrial activity, and metabolism of amino acids, lipids, and sugars, in addition to various disease-related signaling pathways [205].

### 4.8. Pathways Linking Ferroptosis and ER Stress

The ER stress response elicited by ferroptotic agents and its function in cell death is poorly understood. Recent research has shown that the ER stress response brought on by ferroptotic agents is crucial in the interaction between ferroptosis and other forms of cell death. Ferroptotic chemicals trigger a UPR, which then activates the PERK-eIF2α-ATF4-CHOP pathways through ER stress.

The synergistic connection between ferroptosis and apoptosis is mediated by a p53-independent production of PUMA via the CHOP signaling pathway [82,206].

It is not yet understood which specific cell membranes, such as those in the mitochondria, ER, peroxisomes, lysosomes, and plasma membrane, are involved in the lipid peroxidation that takes place during ferroptosis. It is necessary to remove a bisallylic hydrogen atom from polyunsaturated fatty acyl moieties in phospholipids (PUFA-PLs) integrated into lipid bilayers to start lipid peroxidation. This atom is situated between two carbon double bonds. Remarkably, an abundance of PUFA-PLs makes the cell membranes particularly vulnerable to peroxidation, which appears to be the case for neurons.

Glutamate is the main excitatory neurotransmitter in central neurons, and too much extracellular glutamate is enough to block system x_c_^–^ cysteine/glutamate antiporter, causing ferroptosis [207]. The rise in extracellular glutamate has thus been hypothesized to trigger ferroptosis in physiological circumstances such as the pruning of unneeded neurons in the design of brain networks.

Very recently, it was demonstrated that selenoproteins, including glutathione peroxidase 4 (GPX4), are essential proteins in the brain that participate in redox signaling and reduce lipid peroxides, preventing ferroptosis [208]. 

## 5. Adaptive Reticulum Stress and Its Relationship to Ca^2+^ Dynamics

Stress reactions can have both beneficial and detrimental impacts. Adaptive effects influence resilience, whereas maladaptive effects influence the development of disease states and stress-related pathophysiology. The organism shifts from physiology to disease as adaptation wears out and maladaptation takes over. ER Ca^2+^ concentration is essential for sustaining its oxidizing environment and the luminal ATP levels required for chaperones to operate. The ER-sequestered Ca^2+^ maintains protein synthesis in the lumen while acting as a reservoir for the cation to facilitate the stimulus–response coupling in the cytosol. The UPR is triggered when there is an ER Ca^2+^ release that is sufficient to compromise protein processing.

Local luminal Ca^2+^ concentrations and the dynamic Ca^2+^ flux between subcellular compartments are strictly controlled (Figure 2) [209]. Ca^2+^ entrance across the plasmalemma is regulated by concentration gradients and involves voltage- and/or ligand-gated Ca^2+^ channels [210,211,212]. Na^+^/Ca^2+^ antiporter activity and active transport by plasma membrane Ca^2+^ selective ATPases (PMCA) enable Ca^2+^ efflux. Both the extracellular fluid and intracellular storage sites, primarily the ER, provide Ca^2+^ for the cytosolic pool. Extensive evidence supports the key function of ER-releasable Ca^2+^ in stimulus-response signaling [40,210,213,214]. The cation binds to high-affinity Ca^2+^ receptor proteins, including calmodulin, as [Ca^2+^]_i_ in response to a stimulus rises from resting values near 0.1 M, allowing the activation of several enzymatic processes. 

The relative contribution of ER-sequestered Ca^2+^ varies from one kind of cell to another during Ca^2+^ signaling. Different distributions of Ca^2+^ entry and release channels or localized quantities of Ca^2+^-binding proteins may potentially contribute to variations in Ca^2+^ release from different ER regions. It is well recognized that IP3, which is produced in response to hormonal or other stimuli and acts at receptors (IP3Rs) located in both the SER and RER, releases Ca^2+^ that has been sequestered in the ER [212,215]. In contrast to the cation, which tends to move rather slowly from its place of release, this chemical diffuses quickly throughout the cell.

Ryanodine receptors (RyRs), which come in a variety of isoforms and are found in many different cell types, can also release Ca^2+^ from the ER [216].

RyRs are found near the cell membrane and produce an amplified signal known as Ca^2+^-induced Ca^2+^ release (CICR) in response to Ca^2+^ influx through voltage-gated Ca^2+^ channels. Low [Ca^2+^]_i_ stimulates and high [Ca^2+^]_i_ inhibits the release of the cation through either the IP3Rs or RyRs. 

The Ca^2+^-ATPases of the ER membrane, also known as SERCA pumps, which have some similarities with the PMCA, are active and maintain the gradient of Ca^2+^ concentration between the ER lumen and the cytosol.

## 6. The Cerebellar Granule Cells (CGCs) as a Model to Study the Link between ER Stress and Cell Death in Neurons

Ca^2+^ is a crucial second messenger for cell survival and neural function. As mentioned in the previous section, two ER calcium channels, RyRs and IP3Rs, regulate intracellular Ca^2+^ acting as gates for the passage of Ca^2+^ from the ER lumen to the cytosol.

CGCs are widely investigated for the study of neuronal cell death in vitro and in vivo [217,218]. It has been demonstrated that the proliferation, differentiation, and migration of CGCs are significantly influenced by membrane potential-regulated Ca^2+^ signaling via the calcineurin phosphatase (CaN) pathway [219]. Observations on isolated or organotypically cultured CGCs have shown that these neurons develop and become mature in a temporally regulated manner under physiological extracellular K^+^ concentration −[K^+^]_e_, whereas an elevated [K^+^]_e_ blocks their process of differentiation at an immature stage [220,221] (Figure 3).

### 6.1. ER Stress and Apoptosis in CGCs 

An initial study on elevated [K^+^]_e_ cultured CGCs in the presence of thapsigargin (TG), an inhibitor of ER Ca^2+^-ATPase (SERCA) showed that ER Ca^2+^ stores participate in modulating the response of these neurons to stress stimulation [222]. Subsequently, it was demonstrated that, when given to cultured CGCs, Parkinson’s disease mimetic 6-hydroxydopamine (6-OHDA), which induces ER stress and neuronal death, up-regulated the expression of GRP78 and CHOP, cleavage of proCASP12, an ER resident caspase [223], and the phosphorylation of eIF2α [224]. 

Furthermore, in CGC cultures, S-nitrosoglutathione (GSNO), a likely endogenous nitric oxide (NO) reservoir expressed in the cerebellum [225], caused the sustained elevation of [Ca^2+^]_i_ which resulted partially from the depletion of ER stores as a consequence of ER stress, as indicated by the incomplete splicing of the XBP1 mRNA [226]. The same authors demonstrated that the expression of the proapoptotic growth arrest and DNA damage-inducible gene (Gadd153) was upregulated. These observations highlight the connections between neuronal degeneration and the UPR and are of interest because NO-induced neurotoxicity is involved in various misfolded protein neurodegenerative diseases.

In addition, the ER stressor brefeldin A stimulated the ER resident kinase PERK, increased the phosphorylation of the translation initiation protein eIF2α, and elevated the expression and nuclear localization of the transcription factor Gadd153/CHOP in cultured CGCs [227]. In this study, 2-aminoethoxydiphenyl borate (2-APB), an antagonist of IP3R-mediated Ca^2+^ release and ER-targeted BCL-2 reduced the ER stress-induced CGC apoptosis. In the same study, trophic factor-induced apoptosis (removal of serum in the presence of physiological 5 mM [K^+^]_e_) did not involve ER stress and was not prevented by 2-APB or ER-BCL-2. More recently, ER stress was induced in CGCs using tunicamycin and amyloid beta-peptide (Aβ) with a time-dependent increase in CASP12, GRP78, and CHOP [228,229]. Similar results were obtained when the ER stress was induced by thiamine deficiency [230], or when CGCs were exposed to ethanol which drastically enhanced the expression of GRP78, CHOP, ATF4, ATF6, phosphorylated PERK, and eIF2α when ER stress was induced by tunicamycin and XBP1 [231].

**Figure 3 ijms-23-15186-f003:**
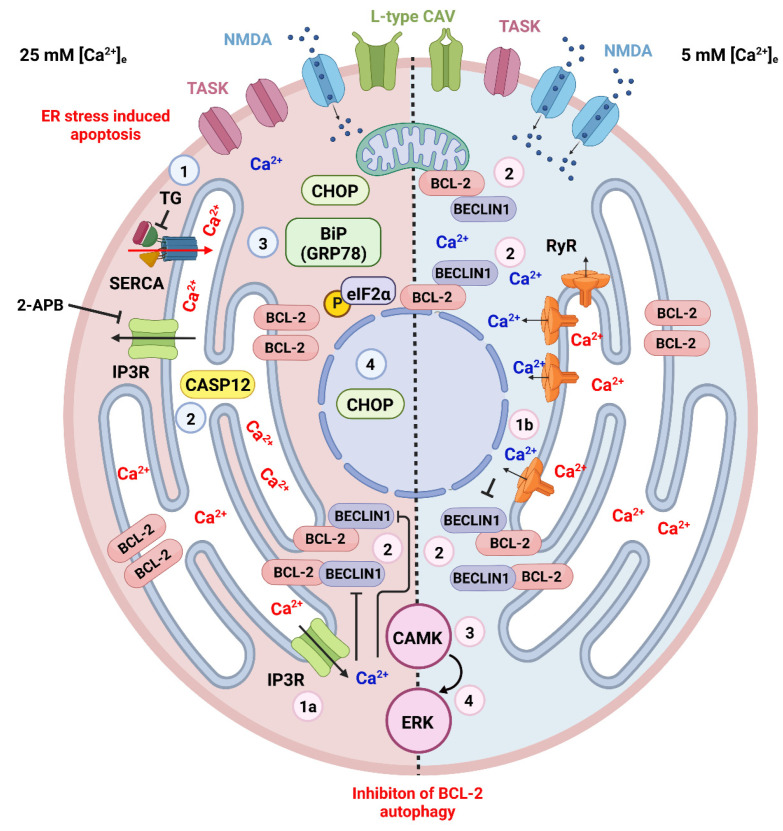
ER stress and cell death in the CGCs. Apoptotic and autophagic mechanisms are dependent on the cytosolic and ER intraluminal concentration of Ca^2+^. These, in turn, are related to the culture conditions in high (on the left represented with pink cytoplasm) or physiological (on the right represented with azure cytoplasm) extracellular K^+^. Different cultural conditions lead to distinct patterns of expression of plasmalemmal receptors, particularly TASK channels that contribute to the resting membrane potential, L-type Ca^2+^ channels that were demonstrated to interact with RyRs in CGCs [232], and NMDA glutamate receptors. Several drugs have been used to induce ER stress in CGCs (see the main text) cultured in 25 mM [K^+^]_e_ eventually leading to apoptosis. The figure shows the cellular events following thapsigargin (TG—numbered azure circles), a blocker of SERCA. With SERCA block, the influx of Ca^2+^ into the ER lumen stops, leading to the activation of CASP12 and the expression of the UPR molecules BiP(GRP78), phosphorylated eIF2α and CHOP, the latter eventually being translocated to the nucleus and promoting the transcription of proapoptotic genes. ER Ca^2+^ controls cellular homeostasis through two different mechanisms mediated by IP3Rs (1a—pink circle) or RyRs (1b—pink circle). The outflow of Ca^2+^ from the ER leads to an increase in the cytoplasmic concentration of the ion with the phosphorylation of BCL-2 and its dissociation from beclin1, thereby inhibiting BCL-2 autophagy through the activation of the CAMK-ERK pathway (pink circles 2–4) [128]. This mechanism operates in both high and physiological [K^+^]_e_. Created with Biorender.com (www.biorender.com, accessed on 27 November 2022).

Using organotypic cultures, we then demonstrated that, at both physiological (5 mM) and elevated (25 mM) [K^+^]_e_, Ca^2+^, as a second messenger, post-translationally regulated the level of BCL-2 expression in mouse CGCs [82]. The protein levels could be increased by two main intracellular pathways converging at the ER (Figure 3). The first, occurring in immature CGCs (cultured at 25 mM [K^+^]_e_), was independent of the extracellular Ca^2+^ but dependent on the release of Ca^2+^ from the ER across IP3R Ca^2+^ channels as demonstrated after blocking these channels with 2-APB. The second ensued in mature CGCs (cultured at 25 mM [K^+^]_e_) was instead dependent on extracellular Ca^2+^ and triggered by a Ca^2+^-induced Ca^2+^ release (CICR) across RyRs expressed at the ER membrane, as it could be blocked by caffeine or dantrolene. Remarkably and in keeping with the observations in [227], the stimulation of IP3-Rs with thimerosal in the presence of physiological [K^+^]_e_ was ineffective, thus confirming that different mechanisms operate in immature and mature CGCs to regulate the ER Ca^2+^ release.

In keeping with our findings, it was reported that the dysfunction of the IP3Rs promoted cell death during ER stress [233]. The heterozygous knockout of brain-dominant type1 IP3R1 resulted in neuronal susceptibility to ER stress in vivo, and IP3R1 knockdown enhanced ER stress-induced extrinsic apoptosis in cultured cells. The IP3R1 tetrameric assembly was positively regulated by the ER chaperone GRP78 in an energy-dependent manner. ER stress provoked IP3R1 dysfunction through an impaired interaction of the two molecules. The authors concluded that IP3R1 senses ER stress through GRP78, resulting in an alteration of the Ca^2+^ signal that promotes neuronal cell death [233].

More recent work has demonstrated that differences in ER stress signaling kinetics determine diverse cell survival outcomes through the activation of mitogen-activated protein kinase phosphatase 1 (MKP-1) in CGCs exposed to tunicamycin or dithiothreitol as stressors [234]. 

Additionally, it was demonstrated that CGCs have a selective susceptibility to N-glycosylation defects compared to cortical neurons, due to these neurons’ inefficient response to ER stress, an important observation considering the existence of congenital disorders of glycosylation [235]. The fact that CGCs may have an inefficient response to ER stress, at least under certain culture conditions, has been suggested in a very recent paper that has analyzed the effects of the reduction in the Ca^2+^ concentration in the ER or cytoplasm, ER stress, and MOMP on the activation of CASP3 to conclude that the decrease in the Ca^2+^ ER concentration resulted in rapid MOMP and strong activation of CASP3 in absence of ER stress [236].

Finally, it is of interest that when the expression of ER stress sensors and their downstream targets in the developing cerebellar cortex was analyzed using immunocytochemistry in histological sections, the activation of PERK and IRE1 was observed in normally developing CGC precursors [237]. A second proliferative pPERK-positive population was also detected in the internal granular layer, and, in general, the density of UPR protein-positive cells was found to decrease significantly when profiles in early and late postnatal ages were compared. 

### 6.2. ER Stress and Autophagy in CGCs

Damaged proteins and organelles can switch metabolically between the ubiquitin-proteasome and autophagy-lysosome pathways [238,239]. Remarkably, [Ca^2+^]_i_ is one of the signals regulating the autophagic cell death [240].

We showed that ER-released [Ca^2+^]_i_ and inhibited BCL-2 autophagy in cultured CGCs, as supported by the observation that when rapamycin inhibits mTOR, autophagy is stimulated, and 25 mM [K^+^]_e_ is ineffective in inducing the overexpression of BCL-2 [127,128]. Autophagy is linked to ERK activation in depolarized rat CGCs [241], however, the mechanisms of neuronal death involving ERK in mouse CGCs may be different as blocking ERK did not interfere with BCL-2 accumulation [127]. This suggests that the sustained inhibition of ERK (and autophagy) reduced survival. Our study rather suggested that [Ca^2+^]_i_ regulates the levels of BCL-2 posttranslationally through the CAMK/ERK pathway, consistent with observations in other neurons [242]. Compared to that in apoptosis, the link between ER stress and autophagy in CGCs has only been poorly investigated. However, CAMKIV limited ER stress and inhibited autophagy in cultured adipocytes [243], and ER stress-induced autophagy was demonstrated in the 293T cell line [244]. Additionally, PC12 cells were protected against H_2_O_2_-induced apoptosis and activated autophagic pathways with the intervention of RyRs and IP3Rs [245].

## 7. Conclusions and Further Perspectives

Much of the focus in research on the role of UPR in brain function has been on disease models [246,247] and has focused on the effects of ER stress on cell death. 

In this review, we first briefly recalled the main data in the literature describing the molecular pathways of ER stress. We then considered the links between ER stress and the different forms of cell death that may affect the cells of the nervous system. We finally thoroughly discussed the relevance of CGCs in modeling the link between ER stress and cell death in neurons.

Recent evidence revealed that the control of protein synthesis operated by the ER is critical in brain development, network connection, and synapse function [248,249]. Remarkably, data from CGCs highlight the role of the organelle in controlling death or survival as a response to adaptive reticulum stress and its relationship to Ca^2+^ dynamics. XBP1 brain expression disruption studies in animals have identified selective changes in gene expression that differ from prior characterizations in other tissues [6]. It is unknown how XBP1s influence specific gene expression patterns in the brain. The major components of the UPR may also have ER stress-independent functions, such as, e.g., aging and cell differentiation [247,250]. It may thus be possible that the pattern of UPR activation differs in different cell types of the nervous system, but we still do not know whether and how this relates to neuronal function and animal behavior.

Nonetheless, we do not know how the distinct components of the UPR are activated during CNS development and if they are involved in determining cell fate and cell differentiation. It is also unclear whether the UPR is necessary for inducing neurogenesis.

To gain a better knowledge of UPR in normal brain function and pathology, the way to go is in a cell type-specific manner. Under this perspective, cultured CGCs may be a useful tool to understand the links between UPR and nerve cell survival or death.

## Figures and Tables

**Figure 1 ijms-23-15186-f001:**
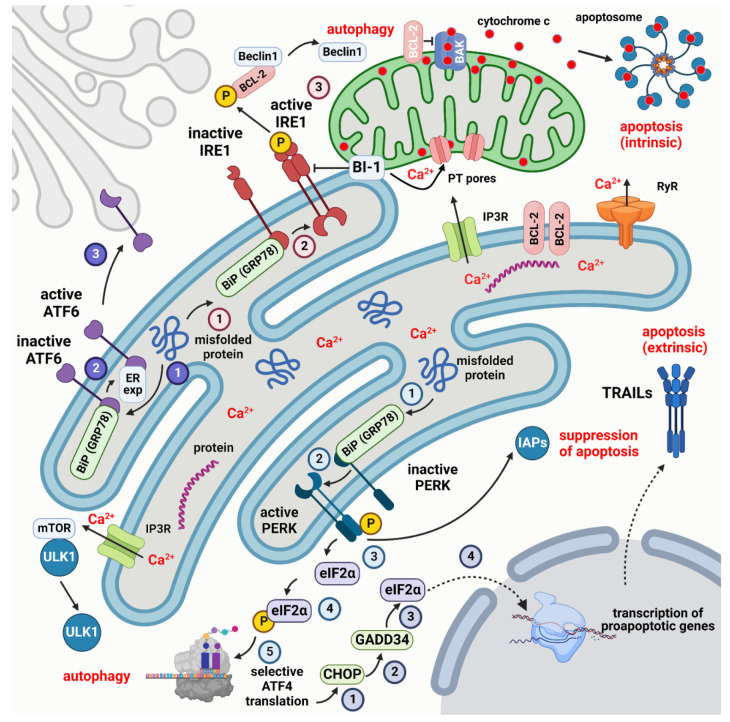
Simplified scheme of the UPR response and its relation to autophagy and apoptosis. The three main sensors that regulate the UPR are PERK, IRE1, and ATF6. Their respective pathways are indicated by numbered circles in light blue, dark blue, and pink. PERK, IRE1, and ATF6 are maintained in an inactive state by binding to the chaperone BiP (GRP78). Misfolded proteins promote the dissociation of BiP (GRP78) and the activation of the three sensors. Active PERK, in turn, phosphorylates eIF2α to start the selective translation of autophagic proteins by ATF4. ATF4 can also activate CHOP that acts on GADD34 to dephosphorylate eIF2α promoting the transcription of proapoptotic genes among which TRAILs that activate extrinsic apoptosis. Active PERK can, however, inhibit apoptosis by binding to IAPs. Once freed from BiP (GRP78), IRE1 dephosphorylates cytoplasmic BCL-2 to release Beclin1 and trigger autophagy. The dissociation of ATF6 from BiP (GRP78) leads to binding to an ER export motif that starts the translocation of ATF6 to the Golgi apparatus. The MAM protein BI-1 controls the Ca^2+^ uptake in mitochondria and intrinsic apoptosis, with the intervention of the pro- (BAK, BAX) and antiapoptotic proteins (BCL-2) of the BCL-2 family. Created with Biorender.com (www.biorender.com, accessed on 27 November 2022).

**Figure 2 ijms-23-15186-f002:**
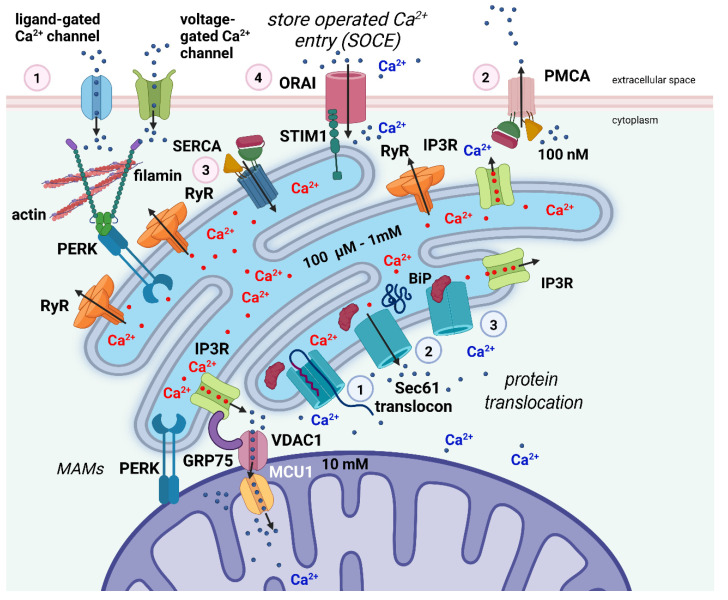
Simplified scheme of Ca^2+^ dynamics inside the cell and its relation to ER stress. In excitable cells such as neurons, Ca^2+^ ions enter the cell through ligand-gated and voltage-gated channels (1—pink circle). The Ca^2+^ gradient across the cell membrane is actively maintained by the PMCA pump (2—pink circle). The ER is the main Ca^2+^-storing organelle. Ca^2+^ is transported against a steep concentration gradient into the lumen of the ER via the SERCA pump (3—pink circle). Upon ligand stimulation, Ca^2+^ ions are released from the reticulum into the cytosol via the IP3Rs and RyRs to activate Ca^2+^-dependent proteins and mediate stimulus-response signaling. The refilling of intracellular Ca^2+^ stores by extracellular Ca^2+^ occurs via SOCE (4—pink circle) by close apposition of the transmembrane proteins STIM1 and ORAI. Filamin A, an actin-binding protein, interacts with PERK independently from the UPR. This interaction seems to be crucial for the formation of juxtapositions of the ER membrane with the plasma membrane, the proximity of the two membranes being a precondition for SOCE to occur. Protein translocation across the ER membrane is mediated via the Sec61 translocon (1—light blue circle). At the end of the translocation process, the Sec61 channel mediates the efflux of Ca^2+^ from the ER to the cytosol resulting in a reduction in [Ca^2+^]_ER_ (2—light blue circle). The ER chaperone BiP binds to the translocon to close it and block the Ca^2+^ efflux from the ER (3-light blue circle). Efficient Ca^2+^ flux from the ER to mitochondria is mediated via IP3Rs, the voltage-dependent anion channel 1 (VDAC1), which is located at the outer mitochondrial membrane, and mitochondrial Ca^2+^ uniporter 1 (MCU1), located at the inner mitochondrial membrane. The IP3Rs–VDAC1 interaction is facilitated by the chaperone GRP75, which is enriched in MAMs. Created with biorender.com (www.biorender.com, accessed on 27 November 2022).

## Data Availability

Not applicable.

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
