# Peer review of "Endoplasmic Reticulum Stress Signaling and Neuronal Cell Death"

_ijms, 2022, doi:10.3390/ijms232315186_

Round 1
Reviewer 1 Report
Merighi et al., summarized ER stress signaling and its role in neuronal cell death. Overall, the manuscript is well-written.
Some comments are listed below:
In section 4, how about the effect of ER stress on ferropotosis?
It is better to describe the effect of ER stress in mammalian and non-mammalian animals individually.
What is the major difference between section 4 and section 5? Maybe they can be merged together.
Although the title is ER stress signaling and neuronal cell death, most part of the review focuses on ER stress. I feel too much emphasis has been put on ER stress. I suggest to curtail the some of very well-known details about ER stress. You should also make a conclusion on what questions need to be addressed beyond what we already know and what is the outlook of ER stress in neuronal cell death research. Therefore, a summary and perspectives need to be added to the end of the review.
Author Response
We warmly thank the reviewer for her/his very useful comments. Below are our point-to-point responses to these comments.
In section 4, how about the effect of ER stress on ferroptosis?
RESPONSE: Two sections dedicated to ferroptosis were added to the revised manuscript (lines 761-795)
It is better to describe the effect of ER stress in mammalian and non-mammalian animals individually.
RESPONSE: We have removed the references to non-mammalian species since they were very recently revised by one of us (ref. 21 in the revised manuscript).
What is the major difference between section 4 and section 5? Maybe they can be merged together.
RESPONSE: We have merged the two sections as suggested.
Although the title is ER stress signaling and neuronal cell death, most part of the review focuses on ER stress. I feel too much emphasis has been put on ER stress. I suggest curtailing the some of very well-known details about ER stress. You should also make a conclusion on what questions need to be addressed beyond what we already know and what is the outlook of ER stress in neuronal cell death research. Therefore, a summary and perspectives need to be added to the end of the review.
RESPONSE: We have shortened the part on the ER stress and added a conclusion as indicated (lines 1009-1037).
Reviewer 2 Report
1) The introduction should be extended. There are several types of cells in the brain. At a minimum, neurons and astrocytes should be characterized, as well as the features of Ca2+ homeostasis in the reticulum of electrically excitable and electrically non-excitable cells. The pathways of cell death in the brain associated with reticulum stress should be briefly characterized.
2) 2.2. Functions of the ER. 4.1. apoptosis. Introductory paragraph needed
3) The phenomenon of adaptive reticulum stress and its relationship to Ca2+ dynamics should be discussed.
4) A separate scheme for the regulation of Ca2+ homeostasis in the reticulum and cytoplasm under stress is needed. In general, little attention has been paid to Ca2+ ions.
5) A brief conclusion or a small discussion of a large analytical review, I also recommended that the authors provide
Author Response
We warmly thank the reviewer for her/his very useful comments. Below are our point-to-point responses to these comments.
1) The introduction should be extended. There are several types of cells in the brain. At a minimum, neurons and astrocytes should be characterized, as well as the features of Ca2+ homeostasis in the reticulum of electrically excitable and electrically non-excitable cells. The pathways of cell death in the brain associated with reticulum stress should be briefly characterized.
RESPONSE: We have extended the introduction as suggested (lines 39-44 and 50-85).
2) 2.2. Functions of the ER. 4.1. apoptosis. Introductory paragraph needed
RESPONSE: We have added an introductory paragraph.
3) The phenomenon of adaptive reticulum stress and its relationship to Ca2+ dynamics should be discussed.
RESPONSE: We have discussed the adaptative reticulum stress in a dedicated new section of the revised manuscript (lines 798-858).
4) A separate scheme for the regulation of Ca2+ homeostasis in the reticulum and cytoplasm under stress is needed. In general, little attention has been paid to Ca2+ ions.
RESPONSE: We have added an ad hoc figure (Figure 2) as requested.
5) A brief conclusion or a small discussion of a large analytical review, I also recommended that the
RESPONSE: We have added a conclusion/perspective discussion as requested.
Round 2
Reviewer 1 Report
My questions have been addressed by the authors.
Reviewer 2 Report
The article has been significantly improved. The authors have taken into account all my comments. The article can be accepted for publication